# Evaluating Participation: Empirical Analysis of Recipient and Beneficiary Engagement with IFAD International Development Projects

**Seokwoo Kim [1], Hyuk-Sang Sohn [2],* and Jinyoung Lee [3],***

1   Department of International Relations, University of Seoul, 163 Seoulsiripdaero, Seoul 02504, Korea; ksw@uos.ac.kr
2   Graduate School of Public Policy & Civic Engagement, Kyung Hee University, 26 Kyungheedaero, Seoul 02447, Korea
3   Center for International Development Cooperation, Kyung Hee University, 26 Kyungheedaero, Seoul 02447, Korea
*   Correspondence: hsohn@khu.ac.kr (H.-S.S.); jeanyoung1127@gmail.com (J.L.);
    Tel.: +82–296–102–023 (H.-S.S.); +82–2961–0590 (J.L.)

**Abstract:** Active participation of the recipient governments and the beneficiaries is an essential factor in carrying out international development projects. Despite ongoing theoretical discussion on the effects of participation by the recipient governments and the beneficiaries in international development projects, there has been relatively little empirical analysis of the effects of their participation in development projects. To fill this gap, this study examines the relationship between the participation of the recipient governments and beneficiaries, and projects outcomes conducted by IFAD by validating two hypotheses. First, the higher financial contribution rate of the recipient governments results in lower evaluation results of international development projects. Second, the higher financial contribution rate of the beneficiaries leads to higher evaluation results of international development projects. In order to verify these two hypotheses, this study analyzed 166 of the IFAD Project Completion Report Validations. We did ordinary least squares (OLS) regression analyses for the panel data made from them. As a result of the analyses, the inverse relationship between the participation of the recipient governments and the outcome of the assessment holds true. On the other hand, the higher involvement of the beneficiaries leads to better results in the assessment. The results reaffirm prior research that suggested that the involvement of the recipient governments has a negative impact on project performance and that the participation of the beneficiaries has a positive impact on the projects performance. This study adopted "financial contributions" as the variable to analyze the participation of the recipient governments and the beneficiaries; since it utilized IFAD data, the research focuses on the agriculture sector in terms of international development cooperation. The applicability of these findings in other areas of international development cooperation therefore to be tested in future research.

**Keywords:** IFAD; recipient governments' participation; beneficiaries' participation; project evaluation

## 1. Introduction

This study examines the relationship between the participation of the recipient governments, the beneficiaries, and project outcomes in international development cooperation. Over the past decades, research on the critical success factor of development projects has shifted from input economics (the Harrod-Domar' theory) to "participation" and "empowerment" [1–3]. In the 1940s and 1950s, Harrod and Domar argued that if labor and capital are combined, stable economic growth is

possible [4,5]. Since the 1980s, there have been community-based approaches that emphasized the participation of the recipient governments and the beneficiaries in international development projects. The key argument for this approach is that the involvement of the beneficiaries in the developing countries has a positive effect on development project performance [6–10].

This study is a theoretical and empirical analysis of how the participation of the recipient governments and the beneficiaries in projects carried out by the International Fund for Agricultural Development (IFAD) affects the results of the project evaluation. The competitive advantages of the participation of the recipient government in international development cooperation projects are (a) expansion of project resources, (b) enhancement of comprehensive development, (c) establishment of a legal and institutional foundation, (d) advancement of effectiveness through cooperation, and (e) implementing sustainable projects [8,11,12]. On the other hand, there are shortcomings such as (a) increased inefficiency, (b) intensified corruption, (c) expanded regulations, (d) escalated conflicts caused by different development philosophies and goals, and (e) increasing transaction costs and project delays [12–14].

Also, the participation of the beneficiaries in the project has the merits of discouraging immoral activity, empowering participants, and making use of local expertise and knowledge [15–18]. On the other hand, the demerits of the involvement of the beneficiaries are the possibility of carrying out self-centered projects; a conflict between donors, implementation agencies, and beneficiaries; and the issue of collective action related to development resources distributions. The theoretical discussion has been laid out, and yet there is relatively little empirical analysis of the effects of the participation of the recipient government and the beneficiaries on development projects. Existing studies with statistical and empirical research on extensive cases are limited [7,8,12]. Accordingly, this study endeavors to bridge the gap between the theoretical discussion and empirical analysis of the involvement of the recipient government and the beneficiaries.

A recent study analyzing the results of the World Bank projects suggested that government participation in the development project operational process in developing countries has a negative impact on the outcome of the project [12,14]. Furthermore, the involvement of government with comparatively low-quality governance systems undermines development effectiveness because of a lack of knowledge and expertise related to international development cooperation projects. In most cases, the recipient governments are demotivated due to the differing objectives of the government and donor agencies. Also, the recipient governments sometimes neither exercise nor understand the proper goal of the projects, and misuse their authority, which may negatively affect the outcome of the project [12]. Also, participation of "too" many actors in funding development projects increases transaction costs, creates project delays, and blurs line accountability, reducing project effectiveness [14].

On the contrary, many studies suggest that the effectiveness of development projects increased with the beneficiaries' involvement in the operational process of the projects. The beneficiaries are better positioned to influence the community directly, and thus their participation plays a crucial role in determining the success or failure of the projects. The participation of the beneficiaries in development projects is divided into "commitment" and "control". Commitment is defined as a contribution to policies on development projects, and control is defined as a means to reflect one's intentions in the planning and operational stage [15–18]. Thus, the effort to elevate the participation of the beneficiaries includes the involvement of one's resources, skills, and endeavors in both the planning and operational phase. These actions will, ultimately, positively affect the outcome of the development project.

The purpose of this study is to empirically verify the results of the recipient government's and the beneficiaries' participation in development projects on a practical level. The analysis materials for this study are the final project reports (validation reports) issued by IFAD. A total of 331 evaluation reports were issued by IFAD from 2000 to 2018. Included in this total are various types of documents, including: Completion evaluation, performance assessments, supervision reports, interim reports, midterm reviews, president reports, and project completion report validations. Among them, we selected 166 Project Completion Report Validations because they showed standard information on

the core variables of this study. The validations are written after the IFAD rural development project is implemented by adopting numerical evaluation in terms of efficiency, relevance, effectiveness, impact, and sustainability. This study applied a statistical data analysis by evaluating 166 project reports, conducted by IFAD between 1998 and 2012, in terms of total project cost, project duration, region, year, the recipient government's participation ratio, and the beneficiaries' participation ratio.

Since every project implemented by IFAD varies in scope, context, and implemented year, there may be a limitation on comparing all the projects to a single standard. However, IFAD provides systematic evaluation data to ensure the validity of the empirical findings. The reliability of the data is also made sure as it is evaluated by the Independent Office of Evaluation (IOE) of IFAD. In order to elevate the reliability and validity of the results, this study verified the collected data using SPSS Statistics (version 25) and compared it with several reports to reaffirm the analysis.

This study has two distinct aspects compared to previous studies. First, the research went beyond assessing the performance analysis of the project cost by the recipient governments; it included the rate of participation of the beneficiaries. Second, this study concentrated on the agriculture sector of the international development cooperation, and the findings have a potential for application to other sectors', which opens up the scope of this study.

This study is structured as follows. It addresses the research question followed by an analysis of previous research on development participants, and reaffirms the significance of participation in the agriculture sector. After that, we describe the characteristics of the collected data and methodology adopted for this study, including an outline of how IFAD conducts project evaluation. We examine the main hypotheses of this study by an interpretation of the analysis results. After discussing these results, the final section provides a summary, implications, and limitations of the findings.

## 2. Literature Review

In carrying out international development cooperation projects, the participation of the recipient government and the beneficiaries is an important issue. There has been growing interest in empirical studies that can demonstrate the relationship between them. This study will review the existing discussions to analyze the results of the projects based on the level of participation of the recipient government and the beneficiaries. Through this review, we attempt to identify key variables in the existing research and to narrow the gap by utilizing empirical data on participation of the beneficiaries, which were not adequately covered by previous studies.

First, this study analyzes the discussions regarding the participation of the recipient government and the beneficiaries for enhancing the effectiveness of international development cooperation, and examines the link with empirical research. In particular, this study will discuss the effectiveness of participation of the recipient government and the beneficiaries, find the relevant empirical analyses, and summarize their key arguments.

Studies of the correlation between the participation of the recipient governments and performance in development cooperation projects, mainly focus on how a difference in governance affects project performance [12,13]. A number of previous studies have demonstrated causal effects between a developing country's governance and aid effectiveness. These studies tends to emphasize the ways in which good governance and policy impacts aid effectiveness [19–24]. These studies directly and indirectly explain the importance of the government's role in the implementation of development cooperation projects. Such studies show that if the governance of the recipient countries is well-organized, there is a high probability that the aid implementation will be transparent [8,11,12]. The existing research focuses on the macro-level approach (governance, institution), which is easier to compare among the recipient countries. However, there is a lack of empirical investigations into how the recipient countries participate and how their level of participation affects the project performance.

Skeptical discussions of the recipient government participation in relation to project outcomes argue that (a) the objectivity of the participation and project evaluation cannot be guaranteed and (b) subjective evaluations will increase the subjectivity of the assessment. Above all, the increase of the

beneficiaries' participation can be misinterpreted as having an impact on project outcomes without objective data, resulting in problems with data collection and bias [8].

Unlike discussions on the participation of the recipient government, studies showing the correlation between the participation of the beneficiaries and the resulting project performance were mainly centered on an in-depth study or participation approach. These studies argue that the higher the participation of the beneficiaries in carrying out international development cooperation projects, the more positive the project's performance [6–10,25]. In particular, they emphasize the importance of the participation of the intended beneficiaries to improve project performance [8,26].

So, what does beneficiary participation mean, and how can it be measured? Some research has shown that the increased level of the beneficiaries' participation contributes to the project's effectiveness and is therefore economically justified [7,8,10]. The reason is that beneficiary participation takes advantage of the indigenous knowledge of the region.

In particular, at the stage of project implementation, the financial contribution of the beneficiaries has a positive effect on project effectiveness and sustainability [7]. Financial contributions from beneficiaries are closely related to the success of small-scale farming and fishery projects [7,27]. Capital investment by the beneficiaries in the form of labor input is a representative form of participation in rural development programs.

Next, there is research that focuses on specific sectors where the participation of the recipient government and the beneficiaries has a positive or negative effect on the project. Research suggests that the participation of beneficiaries has had a positive or limited impact on poverty alleviation in the health, education, and micro-finance sectors [28–30]. The IFAD rural development projects, which are the subject of this study, are development projects focused on farmers, including rural development and income growth.

The participation of beneficiaries in rural development project can have a greater positive effect than in other sectors such as healthcare. Without the participation of the beneficiaries, the goals of rural development projects, including increased agricultural productivity, pest control, construction of villages, and irrigation infrastructure, cannot be achieved. Thus, in this regard, the agricultural sector has aspects that contrast with other development projects. Moreover, the involvement of farmers has a positive effect due to their knowledge, experience, and expertise [7].

After reviewing the prior studies, it is difficult to establish an argument on whether the participation of the recipient governments and beneficiaries has a consistent impact on development projects. Various factors, including the circumstances of the recipient countries, development issues, and beneficiaries' participation, could affect the outcome of international development cooperation. Based on the existing theoretical and empirical studies, however, there may be a distinct relationship between the participation of recipient governments and beneficiaries in the agriculture sector and the outcomes. In other words, the participation of the recipient governments in development projects could have a negative impact. However, the participation of the beneficiaries has opposite effects. Thus, this study sets out two hypotheses based on prior discussions and empirical analysis. In this case, participation is assumed to be the recipient governments' financial contribution ratio and the beneficiaries' financial contribution ratio.

**Hypothesis 1.** *The higher the rate of financial contribution of the governments of the recipient countries, the poorer the results of development cooperation projects.*

**Hypothesis 2.** *The higher the rate of financial contribution of the beneficiaries, the better the results of development cooperation projects.*

## 3. Materials and Methods

This study uses IFAD Project Completion Report Validations data from 1998 to 2012. In each report, the basic project data are included. In the data, there is information on variables used in empirical

analyses of this study, such as region, country, financing type, date of approval, closing date, total project costs, IFAD financial contribution ratio, borrower's financial contribution ratio, and beneficiaries' financial contribution ratio. In addition to these data, the IFAD Independent Office of Evaluation (IOE) provides a dataset for project information and its evaluation results using systematic assessment of development projects. Notably, in evaluation results, the report provides details of results by different criteria including impact, relevance, effectiveness, efficiency, and sustainability, which are standard evaluation criteria suggested by OECD (Organization for Economic Cooperation and Development). Using these consistent and systematic evaluation results, we can estimate the determinants of these evaluation results.

*3.1. Data: IFAD Independent Office of Evaluation Project Completion Report Validation*

3.1.1. Dependent Variable

In this study, we use IFAD IOE's "rating comparison" data for dependent variables in empirical analyses to assess the quantitative effects of the governments' and the beneficiaries' financial contribution ratio on the outcomes of development project performed by IFAD. In the "rating comparison", the IOE provides various evaluation results by various criteria. They include project performance, rural poverty impact, other performance criteria (one of them is sustainability), and overall project achievement. Following the standard evaluation criteria suggested by OECD, this study uses the mean values of impact, effectiveness, and sustainability as a dependent variable in the first set of regression analyses, and the mean values of all five criteria (impact, effectiveness, sustainability, relevance, and efficiency) as the dependent variable in the second set of regression analyses.

Of these five criteria, relevance covers the relevance of objectives, the relevance of project design, and the relevance of targeting. Effectiveness refers to whether the implemented project contributed to the achievement of the objectives of the project. Efficiency covers time-lapse, project costs, cost per beneficiaries, international economic rate of return at project completion, and project management costs. Impact (or rural poverty impact) covers household income and assets, human and social capital, food security, and agricultural productivity, and so on. Sustainability covers agricultural development and technology transfer, savings and cooperatives, engagement of people, innovation and scaling up, gender equality, and so on.

This study uses the IOE rating of each criterion, and the rating scale is as follows: 1 = highly unsatisfactory; 2 = unsatisfactory; 3 = moderately unsatisfactory; 4 = moderately satisfactory; 5 = satisfactory; 6 = highly satisfactory; n. p. = not provided; n. a. = not applicable. By using these rating scales on each criterion, we made two mean values for the dependent variables for regression analyses. If this study had used the original six rating scales of IOE, the appropriate methods of estimating models are those of ordered logit and ordered probit. However, because we use the mean values of rating scales on three and five criteria for each regression analysis, the method applied for estimating models is ordinary least squares (OLS).

3.1.2. Independent Variables

This study has two main independent variables, each of which estimates the quantitative effects of financial contribution of the government and the beneficiaries in each development project. In each development project performed by IFAD, the total project costs consist of an IFAD loan, borrower's contribution, beneficiaries' contribution, and other participants' contributions (or, co-financier, if any, e.g., UNDP). To assess the effects of the borrower's participation and the beneficiaries' participation in the project, this study uses each participant's financial contribution over the total project costs as two main independent variables. Then, the ratio of the borrower's financial contribution/total project costs represents the borrower's participation ratio in the project. At the same time, the ratio of the beneficiaries' financial contribution to the total project costs represents the beneficiaries' participation

ratio in the project. For regression analyses for each dependent variable, this study applies three different models. The first model includes both the main independent variables. In contrast, the second and third model include only one primary independent variable to avoid possible collinearity between the two leading independent variables.

In addition to these two main independent variables, this study includes control variables that may affect the variation of the dependent variables. The control variables included are total project costs, project duration, number of beneficiaries, average annual economic growth rate of the recipient country, GDP per capita of the recipient country at the beginning year of the project, and World Governance Indicators (WGI) values. The WGI consist of six dimensions of voice and accountability, political stability and absence of violence, government effectiveness, regulatory quality, the rule of law, and control of corruption. Based on the existing literature on the relationship between governance and development project outcome, this study includes two dimensions of government effectiveness and control of corruption in empirical analyses. All six control variables are expected to positively correlate to the development project evaluation results. Greater total project costs are expected to contribute to greater development of the recipient region. The longer project duration is expected to have a longer-term positive effect on the development of the recipient region. A larger number of beneficiaries is expected to have a broader positive effect on the development of the recipient region. The higher economic growth rate is expected to give higher satisfaction to people in the region. The rich are expected to be more satisfied with the new development project. A higher score on WGI is expected to lead to a better outcome in a development project performed by IFAD because of the recipient government's greater effectiveness and lower corruption. Out of these six control variables, values for three variables of total project costs, the number of beneficiaries and GDP per capita are log-transformed to reduce heteroscedasticity and the possible methodological problems associated with it.

This study also includes two dummy variables for fixed effects of region and year. The standard method to fix effects in the country-year panel data is to include dummy variables for fixed effects of the country (rather than region) and year. However, this study uses the region dummy rather than the country dummy because the number of countries involved in empirical analyses is vast relative to the total number of cases, and also because many countries have a single development project case. Then, it is impossible to fix the effects of those countries. In the IFAD dataset, geographical regions are divided into Asia, Africa, Europe, and Latin America and the Caribbean. Year dummies are made depending on the beginning of each development project.

## 3.2. Empirical Design

To examine the quantitative effect of borrower's participation and beneficiaries' participation, this study estimated two regression models as follows:

$Y_{it}$ (mean value of evaluation outcome) $= a_{it} + \Sigma \beta_k B_k + e_{it}$

where B1 = recipient government's financial contribution ratio

B2 = beneficiaries' financial contribution ratio

B3 = ln (total project costs)

B4 = project duration (months)

B5 = ln (number of beneficiaries)

B6 = average annual economic growth rate

B7 = ln (GDP per capita)

B8 = mean value of effectiveness and corruption score in WGI index

B9 = region dummy

B10 = year dummy

In the regression models suggested above, *i* and *t* are observations on units (countries) and time points (project beginning years), respectively, $\beta_k$ is a vector of coefficients, and e is the error term.

The first set of regression analyses was done with the dependent variable of the mean value of impact, effectiveness, and sustainability evaluation outcomes. In contrast, the second set of regression analyses is done with the dependent variable of the mean value of five evaluation criteria outcome, including impact, effectiveness, sustainability, relevance, and efficiency. The main reason for this division is that the three criteria of evaluation (impact, effectiveness, and sustainability) are more directly related to real and future development outcomes than the other two criteria of relevance and efficiency. Relevance and efficiency are more about project design and project process, while impact, effectiveness and sustainability are more about project outcome. Because the Pearson correlation coefficient between the two mean values is 0.962, there is no significant difference between the two regression analyses.

The following Table 1 shows descriptive statistics of the included variables.

**Table 1.** Descriptive statistics of variables.

| Name | Variable Description | Obs. | Mean | Min. | Max. | Std. |
|---|---|---|---|---|---|---|
| Outcome 1 | IFAD IOE Outcome Score (mean of impact, effectiveness, and sustainability) | 166 | 3.84 | 0.67 | 5.33 | 0.84 |
| Outcome 2 | Mean of impact, effectiveness, sustainability, relevance, and efficiency | 158 | 3.97 | 2.00 | 5.20 | 0.65 |
| % of borrower | Government's financial contribution ratio | 159 | 17.20 | 0.00 | 89.19 | 16.23 |
| % of beneficiary | Beneficiaries' financial contribution ratio | 137 | 6.82 | 0.00 | 47.36 | 8.03 |
| Total Project Costs | Total project cost (million USD) | 166 | 36.36 | 3.45 | 314.74 | 42.18 |
| Project Duration | Project duration (months) | 165 | 91.41 | 23 | 167 | 25.29 |
| No. of Beneficiaries | Total number of beneficiaries of the project (#) | 160 | 224,334.30 | 201 | 6600,000 | 598,883.26 |
| Avg. Annual Growth Rate | The average annual economic growth rate of the recipient country during the project | 164 | 5.37 | −2.08 | 18.26 | 2.55 |
| GDP per capita | GDP per capita of the recipient country at the beginning year of the project | 163 | 2111.38 | 214.05 | 13,221.42 | 2441.34 |
| WGI | Mean value of effectiveness and corruption in WGI | 166 | −0.5978 | −1.62 | 0.74 | 0.4273 |

Sources: IFAD IOE Project Completion Report Dataset, World Bank (World Development Indicators, World Governance Indicators).

As is well known, according to the Gauss–Markov theorem, the ordinary least squares (OLS) method produces BLUE (best, linear, and unbiased estimators) beta-coefficients in regression analyses, unless its assumptions are severely violated by the nature of the data. In order to reduce the error of possible violations of the OLS assumptions, three methodological processes were conducted in this study. First, the data were examined for multicollinearity by doing bivariate correlation analyses among independent variables. The results of these tests show that there is not a severe multicollinearity problem in the data. The highest bivariate correlation index is 0.399 between the government's financial contribution ratio and the GDP per capita, which implies that wealthier recipient governments contribute more money to the development projects. Secondly, to reduce the possible heteroscedasticity problem, three variables with significant variance (total project costs, number of beneficiaries, and GDP per capita) were transformed to log values. Thirdly, this study applied a statistical method of the LSDV (least squares with dummy variables) to reduce possible problems related to autocorrelation. This study includes region dummies and year dummies in regression analyses. The consequence

of multicollinearity, heteroscedasticity, and autocorrelation is inflated variance, which results in an inefficient estimation of parameters instead of biased estimators (e.g., methodological processes are expected to produce the best and most unbiased estimators).

## 4. Results and Discussion

### 4.1. Results

The results of the estimation of two models of IFAD evaluation outcome are presented in Tables 2 and 3. In each set of regression analyses, this study tried three multiple regressions to investigate whether the government's and the beneficiaries' financial contribution in development projects affect project evaluation outcomes. In terms of beta coefficients' direction and statistical significance, the two models showed very similar results.

**Table 2.** Effects of actors' participation on evaluation outcome (Impact, Effectiveness, Sustainability).

| Dependent Variable Outcome (Mean of 3) | | OLS Models | | |
|---|---|---|---|---|
| | | (1) | (2) | (3) |
| Main variables | % of borrower | −0.008 (0.005) | −0.010 ** (0.004) | |
| | % of beneficiary | 0.015 (0.010) | | 0.020 ** (0.009) |
| Basic variables | In (total project costs) | 0.248 ** (0.104) | 0.275 *** (0.092) | 0.225 ** (0.104) |
| | Project Duration (months) | 0.006 (0.004) | 0.004 (0.003) | 0.004 (0.004) |
| | Ln (Number of beneficiaries) | 0.131 ** (0.051) | 0.128 *** (0.045) | 0.134 *** (0.050) |
| | Avg. annual growth rate (%) | −0.016 (0.036) | −0.029 (0.030) | −0.015 (0.035) |
| | Ln (GDP per capita) | −0.037 (0.125) | 0.001 (0.109) | 0.100 (0.117) |
| | WGI | 0.210 (0.185) | 0.236 (0.172) | 0.179 (0.186) |
| Year dummies | | Yes | Yes | Yes |
| Region dummies | | Yes | Yes | Yes |
| No. of Obs. | | 125 | 148 | 129 |
| Adj. R-square | | 0.149 | 0.193 | 0.130 |

Note: Standard error is indicated in parentheses. ** at the 5% level, *** at the 1% level. Coefficient estimates for various fixed effects dummy are not reported. The dummy variables included in all the models are year (based on the approved fiscal year of the projects) and region (the base category is Asia).

**Table 3.** Effects of actors' participation on evaluation outcome (Impact, Effectiveness, Sustainability, Relevance, Efficiency).

| Dependent Variable Outcome (Mean of 5) | | OLS Models | | |
|---|---|---|---|---|
| | | (4) | (5) | (6) |
| Main variables | % of borrower | −0.008 * (0.004) | −0.008 ** (0.004) | |
| | % of beneficiary | 0.010 (0.008) | | 0.013 * (0.007) |
| Basic variables | In (total project costs) | 0.190 ** (0.083) | 0.214 *** (0.073) | 0.176 ** (0.083) |
| | Project Duration (months) | 0.001 (0.003) | 0.002 (0.003) | 0.00007 (0.003) |
| | Ln (Number of beneficiaries) | 0.112 *** (0.040) | 0.115 *** (0.036) | 0.121 *** (0.039) |
| | Avg. annual growth rate (%) | −0.031 (0.033) | −0.023 (0.028) | −0.037 (0.032) |
| | Ln (GDP per capita) | −0.081 (0.105) | −0.081 (0.091) | −0.136 (0.010) |
| | WGI | 0.275 * (0.164) | 0.252 * (0.152) | 0.271 (0.165) |
| Year dummies | | Yes | Yes | Yes |
| Region dummies | | Yes | Yes | Yes |
| No. Obs. | | 120 | 142 | 124 |
| Adj. R-square | | 0.150 | 0.214 | 0.142 |

Note: Standard errors are indicated in parentheses. * Denotes statistical significance at the 10% level, ** at the 5% level, *** at the 1% level. Coefficient estimates for various fixed effects dummy are not reported. The dummy variables included in all the models are year (based on the approved fiscal year of the projects) and region (the base category is Asia).

The results of these multiple regression analyses confirmed the two hypotheses of this study. The percentage of the borrower, which is the government's financial contribution ratio in the development project, is negatively related to the evaluation outcome, which shows that a higher borrower's financial contribution ratio results in poorer outcomes for the development project.

Meanwhile, the percentage of the beneficiary, which is the beneficiaries' participation ratio in the development project, is positively related to the evaluation outcome, which shows that a higher beneficiaries' financial contribution ratio results in a better outcome for the development project. Beta-coefficients for these two independent variables are statistically significant at the 5% level in models of (2) and (3), and (5), and statistically significant at the 10% level in the model (6) regression result. The empirical results of the participation of the recipient government reconfirm the results of the empirical studies on World Bank development projects.

Out of the six basic control variables, three turned out to be positively and statistically significantly related to the evaluation outcome. A larger amount of total project costs and a larger number of beneficiaries resulted in a better outcome. This means that the more money was spent on a project, and the more people benefited from the project, the higher the evaluation score was on average. Also, a higher WGI resulted in a better outcome in two models. This shows that the recipient government's higher effectiveness and lower corruption when managing domestic affairs contributed to more successful development cooperation projects. These empirical results reconfirm the previous arguments and empirical findings [23,24]. Furthermore, project duration is positively related to evaluation outcome, even though the relationship is not statistically significant. This result is also consistent with the previous empirical finding that there is a negative relationship between project duration and outcome scores in development projects by Asian Development Bank, Japan International Cooperation Agency, GIZ (Deutsche Gesellschaft für Internationale Zusammenarbeit) and the World Bank [24]. This result may be related to project characteristics. For example, difficult agricultural projects might take longer to complete require more preparation and supervision, and also may be more likely to result in the poorer outcome [23]. Contrary to our expectations, the average annual economic growth rate and GDP per capita turned out to be negatively related to the evaluation outcomes in most of the models, but the relationship was not statistically significant.

In addition to the above empirical analysis, we review the reports of the projects with high and low mean values of IFAD IOE outcome score, respectively, to reaffirm the influence on project performance depending on the financial contribution of the recipient government and the beneficiaries.

This study reviewed reports on how the mean value of the IOE outcome score has assessed the financial contribution of the recipient government and the beneficiaries in the reports with maximum and minimum value. Despite the fact that the research was intended to verify the impact of active participation and the role of the beneficiaries on assessment results, it was difficult to locate any reference to the active involvement of the beneficiaries; rather, the collected reports referred to the recipient governments.

The report, which corresponds to the minimum value of Outcome 1 (0.67), referred to the negative evaluation of the role of the recipient government in the "Batha Rural Development Project" implemented in Chad in 2013. The implementing agency for the mentioned project was the Ministry of Agriculture and Irrigation.

> *The government implementing agency was painstakingly slow, even after official effectiveness, which was attained 20 months after approval. The country context was not favorable neither for orderly project implementation.*
>
> *(IFAD Project Completion Report Validation "Batha Rural Development Project" p. 7)*

In this regard, comparable findings were recorded in other assessment reports conducted in Chad. There are cases in which the efficiency of the project dropped due to the delays in the government's project implementation procedures. For instance, the evaluation report on Uganda (outcome score: 5) pointed out inefficiency due to government procedures. In 2014 Project Completion Report Validation on "Vegetable Oil Development Project" in Republic of Uganda, the report points out "-government procedures have caused delays in project implementation, which have reduced its efficiency" (p.13).

We reviewed the project reports, which did not directly mention that the participation of the beneficiaries had a positive impact on the project results. The research, however, emphasized that the participation of the beneficiaries is a core element for achieving the project goals.

> *The participatory approach was commendable as well. The equal involvement of the poor and women in planning and implementation and priority given in wage labor works ensured the most vulnerable groups benefited socially and economically from the project intervention, which was unprecedented in the project area. Participation has also been a key factor for the successful targeting efforts of the project.*
>
> *(IFAD Project Completion Report Validation "Ha Tinh Rural Development Project" p. 10).*

### 4.2. Discussion

The results of empirical analysis of this study shows that participation through financial contribution does not guarantee good results for development projects. Although there are clear sets of reasons for the merits of participation [31], this study suggests that there is a need to divide participation into different categories. This study supports the findings of the previous studies, which argue for a negative relationship between the recipient government's participation and the project outcome, and a positive relationship between the beneficiaries' participation and the project outcome.

As we discussed above, there may be many reasons for the poorer project outcome as a result of increased participation of the government. Governance problems, corruption, bureaucratic inefficiency, conflicts of interests and objectives, project delays, increased transaction costs, blurred accountability, and different levels of commitment in different countries contexts may be the reasons for the lower performance of the development projects. This study suggests that the potential conflicts caused by differences in priorities between the recipient government and the beneficiaries, or disagreements between the donor and the recipient country, may hinder the effectiveness of the rural development project. The central or local governments of the recipient country frequently engage in development cooperation projects to promote their own interests, while penalizing farmers with specific objectives and sometimes close ties with potential rivals in the form of local elites.

Meanwhile, this study shows that the higher involvement of the beneficiaries through monetary contribution in development projects results in a better outcome. Although empirical analysis limited to the financial contribution of beneficiaries, it may lead to a sense of ownership and accountability, capacity building, and other forms of participation by the beneficiaries, such as labor input and participation in decision-making process [7,32]. Beneficiary participation also facilitates the incorporation of local knowledge, skills, and resources in the process of project design, leading to a better outcome [33]. Furthermore, the more financially the beneficiaries contribute to the development project, the more they would strive to raise the return rate of the investment resources, which could have a positive impact on the development projects. Despite some of the criticisms on the beneficiaries' participation in rural development projects such as additional burden, involvement against their will, and incapacity of the beneficiaries to influence the direction of a project, this study implies that the benefits of the beneficiaries' participation through monetary contribution outweigh the demerits, resulting in a better project outcome.

Most importantly, this study shows that the form of a financial partnership with beneficiaries is a key factor that directly facilitates the establishment of cooperative and sustainable development resources. Voluntary participation by the beneficiaries themselves helps to frame the procurement of development resources in ways that foster the sustainability of project outcomes. Financial partnerships with the beneficiaries also further expand the scope of domestic participation as an alternative means of mobilizing resources for development projects.

## 5. Conclusions

This study analyzed the reports provided by the IFAD to verify the two main hypotheses and to evaluate the impact of the financial contribution of the recipient government and the beneficiaries on the outcome of the project. The first hypothesis is that the higher the financial contribution ratio of the recipient government, the lower the evaluation of the results of the development projects. In our empirical research and results, we could confirm that the involvement of recipient governments has a negative effect, as argued in previous studies. Above all, many recipient countries still have undemocratic political systems with chronic corruption. Government participation in development projects under these situations may be implemented in ways that exploit the beneficiaries, which of course decreases the effectiveness of the implemented projects. Specifically, the participation of the government in the agricultural sector, is likely to lower the success rate when the government ignores regional characteristics such as regional development demand and farmers' own interests.

The second hypothesis is that the higher the financial contribution ratio of the beneficiaries, the better the results of the development projects. For empirical verification, this study used the "financial contribution" ratio of the beneficiaries in the development projects as the variable to analyze the effects of the participation of the beneficiaries. Also, this study narrowed the various fields down to the agriculture sector by reviewing IFAD reports. Agriculture has distinctive characteristics compared to other development sectors, as international standards cannot be applied horizontally due to significant local and regional variations in climate, main crops and animals, and methods of cultivation. In specific agricultural environments, some factors such as the experience, knowledge, technology, and intuition of farmers living in the area can have a significant impact on the success of the development project. Thus, the participation of the beneficiaries in the agricultural development projects is essential. Indeed, the participation of the beneficiaries takes various forms of time, effort, financial resources, technology, and knowledge. We reaffirm the importance of the beneficiaries' participation in implementing development cooperation projects and the impact of forming partnerships with the beneficiaries. The active participation of not only the recipient governments but also the beneficiaries who receive assistance in the implementation of development cooperation projects is a major factor affecting the outcome of the project.

The conclusions that the recipient government's participation costs project outcome and that the beneficiaries' participation benefits project outcome are not new ones, but given the scarcity of previous studies that analyzed both participations of the recipient government and the beneficiaries together, this study adds further evidence to support the existing arguments for participation. The original contribution of this study is its focus on the participation of both the government and the beneficiaries in the implementation of the development projects and its analyses of the outcome of the development projects related to different levels of the financial contribution. At the same time, this study also confirms the argument that the recipient government's participation does not have a positive impact on the project argument and to find out conditions under which participation of the recipient government results in better project outcomes. Further theoretical and empirical research is needed to verify this argument and to find the conditions under which the participation of the recipient government results in better project outcomes.

This study also has some limitations. One of the methodological limitations is that this study does not perform in-depth qualitative case studies to assess the complex relationship between different forms of participation and project outcomes. Also, because we could not find comparable data on different forms of participation such as legal and administrative support from the recipient government, and labor, time, knowledge, and expertise from the beneficiaries across the projects, we had to focus only on financial contribution. Due to these limitations, the findings of this study cannot be generalized as the relationship between participation and project outcomes. Instead, this study shows that the financial contribution as one type of participation by the recipient government and the beneficiaries leads to different project outcomes.

In lieu of conclusion, two major implications of this study can be summarized as follows: Firstly, this study has empirically reconfirmed the finding of earlier research claiming that the participation of the recipient governments in development project leads to negative project outcomes. To enhance the effectiveness of the participation of the recipient government, other factors such as government ownership, mutual interests between the recipient government and other participants, a transparent legal system, and policy-making and implementing procedures may be crucial. To verify the conditions for the successful participation of the recipient government in development projects, future research is needed. Additionally, this study demonstrates the importance of the participation of the beneficiaries in the development projects in the agricultural sector. This study found that the greater the financial contribution of the beneficiaries to agricultural development projects, the better the project outcomes. It can be assumed that the participation of the beneficiaries in the development project led to greater contributions of time, knowledge, experience, and expertise to the development project. To see whether these findings can be applied to other development fields, future research is needed.

**Author Contributions:** All three authors collectively made theoretical discussion and corresponding hypotheses. In addition to that, S.K. did statistical analyses and interpretation of empirical results, H.-S.S. wrote introduction and conclusion and secured funding, and J.L. collected raw data, reviewed existing literature and did qualitative analyses. All authors have read and agreed to the published version of the manuscript.

**Funding:** This work was supported by the Ministry of Education of the Republic of Korea and the National Research Foundation of Korea (NRF-2018S1A3A2075117).

**Acknowledgments:** Special thanks to Michael Feener of Kyoto University for his valuable comments on the draft manuscript.

**Conflicts of Interest:** The authors declare no conflict of interest.

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
