# Peer review of "Evaluating Participation: Empirical Analysis of Recipient and Beneficiary Engagement with IFAD International Development Projects"

_sustainability, doi:10.3390/su12145862_

Round 1

Reviewer 1 Report

This paper contributes to a relatively small set of studies that examine project outcome ratings in international development projects.  This literature is small relative to the number of papers that incorporate foreign aid into cross-country growth regressions, and yet these types of studies might very well do more to help us better understand the effectiveness of development aid.  The authors study 148 projects from the International Fund for Agricultural Development and find that projects with more beneficiary involvement receive higher ratings, while projects with more government involvement receive lower ratings.  In both cases, “involvement” is measured as the proportion of total project funds originating with the corresponding actor.

Regarding the hypotheses, the authors do not provide much justification for the first hypothesis that government participation will lower evaluation scores.  I would like to see more theoretical development with regard to why we should have this ex ante expectation.  On the other hand, the material in the paragraph beginning on line 126 perhaps points to a conditional hypothesis (i.e., that the effects of government involvement will vary with the quality of the government), but the authors have not offered or tested this hypothesis.  The existing literature does somewhat more to motivate the second hypothesis.

The authors have used the breakdown of project financing to proxy for levels of government and beneficiary involvement.  They admit on lines 352-353 that it was difficult to find much discussion of beneficiary involvement in the evaluation reports.  I would encourage the authors to look through additional project documentation to see if they can truly justify a correspondence between the financing breakdown and the levels of involvement in projects.

The authors tell us that they do not have enough countries to be able to estimate effects off of within-country variation (258-259).  One major endogeneity concern is a selection issue where IFAD designs projects with more beneficiary participation for countries where the projects would have worked well even without the beneficiary participation.  Studying within-country variation would help to address this problem.  The authors have not provided us with full information about the number of countries in which the projects in the data are found.  As long as there are a non-negligible number of countries with two or more projects, I would strongly encourage the authors to run a fixed-effects model.

As described in Winters’s (2014) International Studies Quarterly article about corruption in World Bank projects, one of the common reasons offered for low ratings of World Bank projects is the fact that the government never provides the counterpart financing that it has committed.  Do the authors find any evidence that this is the explanation for low IFAD project ratings where there is a larger borrower commitment?  If so, this suggests a slightly different story about why projects with high levels of borrower involvement receive low project ratings.

What explains the variation in missing cases?  Why is it sometimes possible to code for borrower contribution (N=159) while not being able to code for beneficiary contribution (N=137)?  This type of data does not typically explicitly state zero values, so if some breakdown of information is given, but beneficiaries are not included, shouldn’t this be considered a zero?

The project duration variable includes surprisingly large values.  According to Table 1, the average project in the data is a seven-and-a-half year project, and there is at least one project in the data that went on for almost 14 years.  These are very long time spans for development projects.  Is the project duration variable including preparatory work?  Are IFAD projects typically longer in duration that projects run by other donors?  Briggs (2019) in the Review of International Organization studies projects from seven different multilateral donors, and the average project length in the data is 3.5 years – less than half of what the current authors are reporting.  The Briggs data includes IFAD, so I really am surprised by the values that the variable is taking on.

Table 4 repeats information already found in Table 1.

I found the paragraph beginning on line 95 particularly hard to understand.

The citation style seems very odd: is it really the style of Sustainability that the same citation is given new numbers over and over?

The Briggs (2019) article mentioned above and Denizer, Kauffman, and Kraay (2013) from the Journal of Development Economics seem like important project-level aid effectiveness studies that are omitted from the authors’ review of the literature.

Reviewer 2 Report

This is an interesting article that focused on the influence of participation in development outcomes, especially evaluation outcomes. The article provides some important insights into the literature related to participatory development and empowerment. The article has been organized well, flowed from one idea to another. However, the authors need to pay attention to some necessary improvements.

The ratio of participation is considered based on monetary value, and there is no indication of how other contribution e.g. time, knowledge, skills etc. play a role. The ratio of participation is simplistic and needs further clarification. Without such clarification, the conclusion or relating the findings to the broader literature of participatory development is problematic. Also, it is not clear why t two models were run (outcome variables), one with three dependent variables and another with five variables.

Discussion and conclusion need improvement for connecting the findings to the broader body of knowledge of participatory development. At this point, it reads very simplistically, without considering critical discussion and connecting the findings to the recent and groundbreaking literature.

The section interpretation is confusing--perhaps it should be the results/findings of your analysis and then a discussion/interpretation of findings after the result is needed. This is currently missing.

Reviewer 3 Report

This manuscript analyzes the relation between economic participation in development projects by local governments and beneficiaries, and the evaluation obtained by those projects. It tries to prove that local government participation has a negative impact on the outcomes of the projects and that the beneficiaries participation has a positive effect. The study verifies that hypothesis by performing statistical analysis on data from projects carried out between 1098 and 2012 by the IFAD.

General comments:

This manuscript analyses an interesting matter, and the hypothesis are relevant and well defined, providing an advance in current knowledge. The methodology seems technically sound, and the authors offer sufficient details. However, I have some general questions:

  1. I would suggest to include in the abstract some information about the methodology used for the analysis.

  1. Also, in the abstract, you stay: "This study adopted 'financial contributions' as one of the variables to analyze participation of the recipient governments and the beneficiaries, and since it utilized IFAD data, the research focuses on the agriculture sector in the international development cooperation.". But when reading the manuscript, I understand that "financial Contributions" is the only variable used to characterize the participation of the recipient governments. Is that right? In that case, I would suggest to explicitly state in the hypothesis that authors are analyzing the economic or financial participation of local governments and beneficiaries. Other variables should be considered if an analysis of overall participation wants to be addressed as you acknowledge at footnote 4.
  2. Other aspects that could be clarified are the implementation mechanism. Did IFAD always provide the funding for the projects through similar tools? I mean, a grant o loan, with similar requirements or characteristics, with comparable monitoring strategies… Was the government participation a requirement for getting the funding? Who was the institution or agency responsible for carrying out the projects? Was it the same in all the projects? Was always a public agency dependent on the local government and that's why the term borrower is used?
  3. Have you analyzed if the results vary depending on the region (Asia, Africa or Latin America)? Is the distribution of countries by region omparable in the sample? Do you have considered if having several samples of the same country could be biasing the results?  
  4. Have you analyzed if the results vary with time?
  5. Results are significant, although I consider that the conclusions section present some affirmations that are not directly obtained from the analysis. For example, it says: "This study implies that potential conflicts caused by the differing objectives between the recipient government and the beneficiaries or disagreements between the donor country and the recipient country can hinder the effectiveness of the project". Or "Central or local governments of the recipient country often implement development cooperation projects to promote their own interests while penalizing farmers with close ties with potential rivals in the form of local elites". The variables you analyze in the manuscript are not related to those statements. Therefore, I would suggest moving this part to a discussion section, where potential explanations for the results can be analyzed, but not considered a direct conclusion from the study.
  6. Although the English language and style are generally correct, a minor spell check is required.
  • Specific comments
  1. At footnote 2, I think that sustainability, relevance and efficiency are missing in the second parenthesis.
  2. Line 205: "Following the standard evaluation criteria suggested by OECD" could you please provide a reference for those criteria?
  3. Line 155: I would suggest saying "have a positive/ negative effect" than "how positive/ negative effect."
  4. Line 166: "agriculture sector has contrasted aspects from other development projects" or maybe "the agricultural sector has aspects that contrast with other development projects."
  5. Line 173: "based on the existing theoretical, empirical studies" or maybe "based on the existing theoretical and empirical studies".
  6. Line 262 you refer to Latina America as Latin. Is that an error or is the term used inside IFAD?
  7. Several references are repeated in the Reference section.

Round 2

Reviewer 2 Report

Thank you for revising the paper.

It is good to notice that the authors acknowledge the limitation of the method and design of the study.

"However, we could not find any information on labor, time input, knowledge and expertise input, and others which may show different types of government and beneficiary participation in the project."

I suggest that you clearly describe the limitations of the method and/or generalization of the study.

The paper did not properly address review point 3 and 4. The above limitations can be critically discussed and the originality and contribution of the findings should be clearly distilled in the discussion/conclusion section.

Otherwise, the manuscript does not offer much value to the readers of the discipline. Please note that it has already known that the participation of both government and clients for a better outcome of the development project.

Reviewer 3 Report

This manuscript analyzes the relation between economic participation in development projects by local governments and beneficiaries, and the evaluation obtained by those projects. It tries to prove that local government participation has a negative impact on the outcomes of the projects and that the beneficiaries participation has a positive effect. The study verifies that hypothesis by performing statistical analysis on data from projects carried out between 1098 and 2012 by the IFAD.

General comments:

The authors' revision has improved the manuscript and the authors' responses were helpful, but I still have concerns about some points:

1. It would be very interesting to have information on how WGI impact on project outcomes and how it relates to government participation. This would help to better explain the impact of government participation. Given that the authors already have the
WGI variable, this analysis seems feasible in this research and could be very important to improve the financial participation of governments in development projects. In fact, it is something that you could have identified as an essential aspect because in your methodology because in the literature review you say: "There are a number of 149 previous studies that demonstrate causal effects between a developing country's governance and aid effectiveness. These studies tends to emphasize the ways in which good governance and policy impacts aid effectiveness [19-24]. These studies directly and indirectly explain the importance of the government's role in the implementation of development cooperation projects".

2. Authors have modified the hypothesis to clarify that they are analyzing the relation between "financial contribution" (of both governments and beneficiaries) and project outcomes. However, throughout the document, it is not clear if authors want to identify "financial contribution" with overall participation without considering other variables (such as experience, knowledge, technology…). Authors state in the manuscript that other variables should be considered in future research, and I understand that. But then the scope of this paper should be clearly limited to "financial contributions" and that terms should be used throughout the paper. Another alternative would be to clearly state in the methodology that participation will be modelled with the "financial contribution" variable, and the lack of other variables should be discussed as one of the limitations of this study. For example, in line 489 you say "At the same time, this study also confirms the argument that the recipient government's participation does not have a positive impact on the project." From my point of view, this study shows that the financial contribution of government does not have a positive impact on the project, and this contributes to reinforce the argument that the recipient government's participation does not have a positive effect on the project. The difference may be subtle but, as you are discussing a quite important issue, I think that those details are relevant.

3. At table 1, should we understand that you have used 159 projects to study hypothesis 1, and 137 projects to explore hypothesis 2?

4. From my point of view, it is essential to differentiate what conclusions can be directly obtained from your research and what explanations or interpretations can be proposed for your results. For example, in line 438 you affirm "This study implies that the potential conflicts caused by differing objectives between the recipient government and the beneficiaries, or disagreements between the donor and the recipient country, may hinder the effectiveness of the rural development project". I do not see how you can justify that affirmation with your results. It seems instead a plausible interpretation, and it should be clearly differentiated. Even if you have shown a paragraph from a report that identifies that relationship, it is only a sample, and it is not valid to establish a scientific conclusion. The potential interpretations could be part of a Discussion section.

In line 505 you say" It can be assumed that the participation of the beneficiaries in the development project led to greater contributions of time, knowledge, experience, and expertise of the beneficiaries to the development project." It is the first time you propose this assumption, and it is not supported by the data. It seems still reasonable, and it could be included in a Discussion section, but not as a conclusion of your research.

In line 379 you state: "Furthermore, project duration is also positively related to evaluation outcome, even though the relationship was not statistically significant.is not statistically significant. This result also reconfirms the previous empirical finding that there is a negative relationship between project duration and outcome scores in development projects […]". It seems a contradiction to say that the results are not statistically significant but that they confirm previous findings.

  • Specific comments
  1. I would suggest using more usual terminology for the sections:
    • Materials and Methods instead of "Data and Empirical Design."
    • Results instead of "Interpretation"
  2. It would also be useful to have a discussion section different from the results section, where the interpretation of the data can be discussed.
  3. In line 34 you say "This study adopted" financial contributions" as one of the variables to analyze the participation of the recipient governments and the beneficiaries," but I would suggest "This study adopted" financial contributions" as the variable to analyze the participation of the recipient governments and the beneficiaries,"
  4. In line 111 and 112, one sentence seems to be repeated.
  5. If, as authors have explained in their responses, the financial mechanism used by IFAC are always loans, then in line 264 I would suggest replacing "grant/loan" by only "loan".
  6. In line 489, the two last sentences seem repeated.

Round 3

Reviewer 2 Report

The author's revisions have improved the paper substantially. The comments have been addressed to my satisfaction. I would suggest some text editing related to sentence structure and phrasing.